# Nanomaterial-Based Label-Free Electrochemical Aptasensors for the Detection of Thrombin

**DOI:** 10.3390/bios12040253

**Published:** 2022-04-16

**Authors:** Hibba Yousef, Yang Liu, Lianxi Zheng

**Affiliations:** 1Department of Biomedical Engineering, Khalifa University, Abu Dhabi 127788, United Arab Emirates; 100059936@ku.ac.ae; 2College of Science and Engineering, James Cook University, Townsville, QLD 4811, Australia; yang.liu11@jcu.edu.au; 3Department of Mechanical Engineering, Khalifa University, Abu Dhabi 127788, United Arab Emirates

**Keywords:** label free, thrombin, aptasensor, electrochemical, nanomaterial

## Abstract

Thrombin plays a central role in hemostasis and its imbalances in coagulation can lead to various pathologies. It is of clinical significance to develop a fast and accurate method for the quantitative detection of thrombin. Electrochemical aptasensors have the capability of combining the specific selectivity from aptamers with the extraordinary sensitivity from electrochemical techniques and thus have attracted considerable attention for the trace-level detection of thrombin. Nanomaterials and nanostructures can further enhance the performance of thrombin aptasensors to achieve high sensitivity, selectivity, and antifouling functions. In highlighting these material merits and their impacts on sensor performance, this paper reviews the most recent advances in label-free electrochemical aptasensors for thrombin detection, with an emphasis on nanomaterials and nanostructures utilized in sensor design and fabrication. The performance, advantages, and limitations of those aptasensors are summarized and compared according to their material structures and compositions.

## 1. Introduction

Thrombin is a serine protease involved in hemostasis [1], converting the precursor fibrinogen into fibrin for the formation of blood clots [2]. Thrombin is not present in the blood under normal conditions, but rather its precursor, prothrombin, can be found in circulation [3]. Upon exposure of tissue factor to blood, the coagulation cascade is activated to generate thrombin from prothrombin at the site of injury, occurring in a precise and balanced manner [4]. Thrombin plays a central role in hemostasis as it converts fibrinogen into fibrin, which then is deposited to entrap aggregated platelets and stabilize the blood clot [5]. Imbalances in coagulation, including the overexpression of thrombin, can lead to various pathologies [2] such as heart attack, stroke, liver disease, deep vein thrombosis, leukemia, and pulmonary embolism, among many more [6,7]. It is reported that COVID-19 patients have elevated thrombin levels on diagnosis [8], involved in a condition known as COVID-19-associated coagulopathy (CoAC), which manifests as numerous thromboembolic complications in COVID-19 patients [9]. Consequentially, it is of clinical significance to develop a fast and accurate method for the quantitative detection of thrombin [2].

In general, the detection of any specific bio-molecule needs a biological recognition element that has a strong affinity to the target [10]. Antibodies have typically been employed as a biorecognition element due to their high affinity to a large range of molecules. However, antibody-based biosensors have suffered from low sensitivity and have therefore not been widely adopted given the fact that biomarkers of most diseases are present only in trace amounts in the bloodstream. Although immunosensors with improved sensitivity have been developed, their tedious and complicated fabrication processes, as well as their enhanced cost, have impeded their practical applications [11].

Electrochemical aptasensors, which benefit from the extraordinary selectivity of aptamers and the high sensitivity of the cost-effective electrochemical techniques, have attracted considerable interest since the pioneering work on protein detection was reported by Kazunori et al. [12]. Nucleic acids (NAs), typically known for their storage and translation of genetic materials, have the potential to perform a more diverse range of functions, including the use as aptamers with a remarkable ability to bind their targets, for the construction of electrochemical aptasensors [13]. Aptamers offer numerous advantages over antibodies as a biorecognition element for biosensing, particularly in terms of their simplicity and low production cost without the requirement of cell or animal lines [14]. Additionally, aptamers can bind to a wide variety of molecules [15], including targets for which antibodies are difficult to obtain [16]. Aptamers can also be easily modified with functional groups [17], whereas this functionalization process in antibodies is not site-specific, which could result in the alteration of target-binding areas, thus interfering with their activity [18]. Furthermore, aptamers are superior in thermal and conformational stability, being able to recover their original configurations even after denaturation [18]. Among various analytical methods, electrochemical techniques are attractive when applied to aptasensing due to the high sensitivity, fast response, small sample consumption, and low cost. In addition, the capability of electrochemical aptasensors in miniaturization and mass production makes them promising for point-of-care applications [19,20,21].

Although many electrochemical aptasensors for thrombin detection have been proposed, there are still some limitations that need to be overcome before the transformation into practical applications. Biofouling of the electrodes is the first challenge when an electrochemical aptasensor is used to detect thrombin in whole blood samples. Blood clotting cascades are initiated upon direct contact of the electrode with whole blood, as it is recognized as a foreign body. This results in the biofouling of electrodes with fibrin, platelets, and blood cells, producing a strong false positive, even in the absence of thrombin in the sample [22]. Therefore, current detection is mainly done through isolated serum samples. Another challenge is the surface modification of the working electrodes with self-assembled monolayers (SAMs). Though it is a well-established method and is the most widely used technique in the immobilization of aptamers, the organic nature of SAMs can insulate the electrode and hinder electron transfer [23]. Lastly, in the case of pathology, thrombin is present in the blood at nM levels, therefore requiring highly sensitive biosensors for successful detection [6].

These issues and problems could be theoretically resolved by utilizing nanomaterials and nanostructures. Nanomaterials could be used to design and fabricate ultramicroelectrodes (UMEs) [24] which offers a small device size [25], improved sensitivity, and a high signal-to-noise ratio. Many nanomaterials could be used in a variety of formats; they have several advantageous properties, including biocompatibility, structural stability, and good electrical properties [26]; more importantly, they can be easily bio-functionalized [27]. Combined with their large surface area, they provide a platform for increasing the density of immobilized aptamers, thus enhancing their sensitivity [28]. Many nanomaterials have excellent conductivity and intrinsic catalytic activity and can therefore act as redox probes without the requirement of additional costly labeling methods [28,29,30,31]. In emphasizing these material merits and their impact on resolving the above-mentioned challenges in thrombin detection, this paper reviews the most recent advances in label-free electrochemical aptasensors for the detection of thrombin, with an emphasis on nanomaterials and nanostructures utilized in sensor design and fabrication. The performance and advantages/disadvantages of sensors are then summarized and compared according to the material/structure categories after a short discussion on the sensing principle and aptamer immobilization. The challenges in the area and the scopes of this review are schematically illustrated in Figure 1.

## 2. Sensor Structure and Sensing Principle

Electrochemical aptasensors consist of two main components, the aptamer that serves as the bio-recognition element and the transducer that works to convert the aptamer binding events into a measurable signal [32]. The transducer can be further described as two main constituents: the electrode system and the electronic readout unit. Signal generation is based on oxidation-reduction reactions, mediated by redox-active molecules that are covalently bound to the aptamer, such as enzymes, catalysts, or nanoparticles. Alternatively, redox mediators can be present in the solution, allowing for electron transfer through diffusion to the electrode surface [33].

The electrode, which undergoes surface modifications, is used as a platform for the immobilization of aptamers [34]. Upon binding to a specific analyte, the aptamer incurs conformational changes which affect the relationship between the redox probe and the electrode, such as reducing/increasing the electron transfer distance, changing their electrostatic interaction, desorbing the redox probe or molecule from the aptamer bases, or blocking the access of probe to the electrode, etc., as shown in Figure 2a–d. This, in turn, generates a signal that is quantitatively associated with the amount of the analyte in a sample [18]. The signal generation and detection can be conducted using several electrochemical techniques such as electrochemical impedance spectroscopy (EIS), cyclic voltammetry (CV), differential pulse voltammetry (DPV), and square wave voltammetry (SWV). The EIS detection relies on the change in the resistance of charge transfer (R_ct_) at the electrode surface due to the interaction between the immobilized aptamers and their targets, which can be probed in the presence of reversible redox couples. The CV analysis is based on the change in current response of the electroactive molecules that are associated with the specific binding of aptamers and targets, while the sensitivities could be limited by the background currents that arise from the double-layer capacitance at the electrode surface. As compared with CV, the DPV and SWV techniques are more widely used for the development of aptasensors because higher sensitivities and lower limits of detection could be achieved, benefiting from the improved signal sampling approaches to eliminate background noises. In addition, the combination of various electrochemical techniques for aptasensor characterization and fabrication has been highly attractive for the understanding of sensing mechanisms, which is of great importance for optimizing the sensing parameters and enhancing the analytical performance, such as sensitivity, selectivity, and stability. Regarding electrode materials, gold and carbon are the most commonly used due to their unique properties, including good conductivity and chemical stability. Gold electrodes exhibit good biocompatibility and low electrical resistance [35], though their large popularity in aptasensors is mainly due to their ability to immobilize thiol-modified aptamers by self-assembly [36]. Glassy carbon is an attractive material to construct an inert electrode, owing to its low oxidation rate, small pores, and chemical inertness [37]. Though not as common, indium tin oxide (ITO) has also been utilized [38].

Traditionally employed labeled-antibody biosensing requires labels, such as enzymes and fluorescent or radioactive molecules to the targeted analyte, and thus is expensive, time-consuming, and frequently results in a reduced affinity of the analyte due to the non-site selective labeling [32,42]. In contrast, the electrochemical aptamer-based detection of thrombin could be label-free. The inherent nature of the redox-active nucleotide bases Guanine and Adenosine can be directly utilized in sensing by monitoring their oxidation reactions [43]. Or, more commonly, redox probes can be attached to the aptamer rather than the analyte via weak interactions such as intercalation or electrostatic bonding, by which the binding site and affinity of the analyte remain un-modified. Upon binding of the analyte to the aptamer, the redox probe is displaced or released due to the aptamer conformation, leading to a change in the electrical signal being measured [18] (Figure 2a,b). Alternatively, redox probes may not be associated with the aptamer, but rather present in the solution, generating the measured signal by diffusing to the electrode [18]. The polyanionic nature of DNA can attract redox-active cations, which will be displaced upon the binding of aptamers to thrombin, resulting in a signal output (Figure 2c) [44]. Finally, the binding of thrombin to immobilized aptamers can act as a barrier that hinders electron transfer between the electrode surface and the electrolyte, resulting in an enhanced current difference (signal off mode) (Figure 2d) [45].

## 3. Structure and Immobilization of Aptamers

### 3.1. Structure and Function of Thrombin Aptamers

Aptamers are synthetically produced via a process known as systematic evolution of ligands by exponential enrichment (SELEX) [46]. The process was initially reported in 1990 [47,48] and has since been used to generate thousands of aptamers against a wide variety of target molecules [49,50]. SELEX is an iterative process, involving multiple repetitive steps [51]. Libraries containing over 10^15^ random oligonucleotide sequences are screened for specific sequences with a high affinity to a determined analyte. These sequences are then amplified via reverse transcription–polymerase chain reaction (PCR), allowing for the domination of the sequences with the highest affinity [52]. The affinity between an aptamer and its analyte is characterized by a value known as the dissociation constant (K_d_), with lower K_d_ values reflective of stronger binding [13]. Iterative cycles of selection and amplification are carried out in order to narrow down the best candidate for the specified analyte. Subsequently, an analyte-specific aptamer is identified [52].

The first thrombin aptamer was isolated in 1992 by Bock et al. [53], which was the first single-stranded DNA oligonucleotide considered for use as an aptamer. This 15 nucleotide sequence (5′- GGTTGGTGTGGTTGG-3′) has been extensively studied as an anticoagulant therapeutic agent and is commonly known as the thrombin-binding aptamer (TBA) [7]. Two aptamers for α thrombin have been synthesized, Apt29 and Apt15, consisting of 29 and 15 nucleotides, respectively [54]. These non-B DNA sequences are able to form secondary, non-canonical structures and do not follow Watson-Crick base pairing [55]. Rich in the nucleotide Guanine, structures, known as G-quartets, are assembled in which four guanine nucleotides form a planar structure via Hoogsteen hydrogen bonds, stabilized at their center by a monovalent or bivalent cation which acts to partially reduce the repulsion between the negatively charged nucleic acids [56,57,58]. G-quartets can further assemble into antiparallel G-quadruplexes (G4) by forming π-π bonds between quartets in a stack-like formation [59]. Antiparallel G4 is a highly organized, chair-like structure, which is the signature structure of TBAs [60]. The G-quadruplex structure of Apt15 and Apt29 is shown in Figure 3a.

As shown in Figure 3b, thrombin contains two positively charged binding sites, known as exosite I (Fibrinogen-binding) and exosite II (Heparin-binding), to which negatively charged complementary regions of the aptamers will bind. Apt29 binds to thrombin at its heparin-binding exosite, whereas Apt15 binds to the fibrinogen-binding exosite, mediated by van der Waals forces and hydrogen bonds [3,62]. Upon binding to thrombin, the aptamer will undergo conformational changes via three-dimensional intramolecular folding, converting from a hairpin to tertiary G4 structure, which is stabilized by thrombin [63].

The presence of two different binding sites in thrombin provides a unique advantage in its detection. This allows the employment of a technique known as a sandwich assay, as shown in Figure 3c, in which two aptamers can simultaneously bind to thrombin, therefore enhancing the detection sensitivity [64]. Traditionally, aptamer-antibody assays have been used in the sandwich format. However, problems such as antibody instability and the high cost of this technique have been resolved by the use of aptamer-pair based detection in its place [64]. Very few kinds of proteins other than thrombin have the advantage of two aptamer binding sites [65].

### 3.2. Immobilization of Aptamers

Immobilization of aptamers onto the surface of the electrode is an essential step in the construction of an electrochemical aptasensor [66], where maximizing surface density and maintaining the binding function of the aptamers are primary concerns [23]. The simplest method of immobilization is the self-assembly of thiol-modified aptamers onto gold electrodes through covalent bonding [67], which leads to the formation of SAMs on the electrode surface [68], as depicted in Figure 4a. Other methods of immobilization include surface modification of the electrode along with the addition of a functional group to the 3′ or 5′ end of the aptamers [66]. The availability of chemical groups on the surface of the modified electrode, such as hydroxyl or carboxyl groups, allows for covalent interaction with an amino-modified aptamer [67], as an example, in Figure 4b. In general, physical adsorption is not recommended, as desorption of aptamers from the surface leads to instability [41]. Further simplified immobilization can be achieved by modifying the electrode surface with carbon nanotubes (CNT) [23] or graphene [69], eliminating the need for functional group modification of aptamers, as these materials can bind directly to aptamers through π-π stacking, as shown in Figure 4c.

Nonspecific adsorption of proteins to the electrode surface could cause a biofouling problem and also have a dramatic effect on the folding and orientation of immobilized aptamers, and should therefore be minimized [36]. Bovine serum albumin (BSA) [45] or mercapto-hexanol (MCH) can be used to mitigate such problems by blocking the surface that has not been occupied by aptamers [67], as shown in Figure 3a. In addition, hydrophilic materials have been used to increase the hydrophilicity of the electrode surface, further increasing resistance to biofouling [71]. More specific details will be discussed in the following sections according to the electrode materials used in aptasensors.

## 4. Nanomaterial-Based Thrombin Electrochemical Aptasensors

### 4.1. Low Dimensional Metallic Nanomaterial-Based Thrombin Electrochemical Aptasensors

Metal-related or metallic nanomaterials are characterized by their superior electrical property, electrocatalytic function, and large surface area [72,73]. From noble metal materials such as gold and silver, to two-dimensional transition-metal dichalcogenides (TMDCs), they have been incorporated into different aspects of electrochemical aptasensors for the detection of thrombin. Here, we discuss them according to their dimensions/morphologies.

Nanoparticles (NPs) are zero-dimensional nanomaterials, possessing different properties compared to the same material on the macro-scale due to quantum effects [74]. Noble metal NPs offer numerous advantages in conductivity, immobilization, and stability for biosensing. Stable immobilization of aptamers is obtained due to the dimensional similarity between the NPs and aptamer, and the large surface area of NPs allows denser immobilization. Electron transfer, especially, is largely enhanced due to the superior conductivity of noble metal NPs [75].

Au NPs are the most commonly utilized noble metal NPs in biosensing, owing to their biocompatibility and electronic properties [27]. Their working principle can be explained by a simple design [76], shown in Figure 5. Au NPs are mainly utilized for TBA immobilizing. TBAs could yield conformational changes upon binding to thrombin, and thus vary the charge transfer resistances in the presence of a redox couple which could be further measured by electrochemical techniques. Such a principle could be used in the sandwich assay design of aptasensors. A recent study by Chen et al. [77] exclusively incorporated Au NPs without additional nanomaterials or enzymes for signal amplification. TBA1 was bound directly to a gold electrode, and TBA2 was immobilized on Au NPs to produce an AuNP-TBA conjugate. The signal was generated using [Ru(NH_3_)_6_]^3+^, as these cations were attracted to the negatively charged backbone of the aptamer DNA sequence. In the presence of thrombin, TBA2 was brought to the electrode surface, allowing for signal amplification by increasing cation adsorption through electrostatic interactions. Despite the simplicity of the aptasensor, DPV measurements revealed a limit of detection (LOD) as low as 0.1429 fM, owing to the electrical activity of the abundant cations transported to the electrode surface through TBA2, thus counteracting the increased electrical resistance brought about by aptamer immobilization. Therefore, this biosensor was highly successful in eliminating additional fabrication procedures by exploiting the inherent electrostatic nature of aptamers and easing the immobilization by utilizing Au-S bonding.

Other noble metal NPs, including silver nanoparticles (Ag NPs), have been utilized in electrochemical aptamers for thrombin detection. Described by Xu et al. [78], Ag NPs were incorporated into a self-polymerizing matrix of dopamine and modified onto a glassy carbon electrode (GCE) for thrombin detection by the EIS technique. Ag NPs deposited on the surface of the electrode showed an increase in current, observed by the CV method when characterizing the electrode surface. Despite the poor conductivity of polydopamine (as evidenced by increased electron transfer resistance in EIS), the hydrophilic properties of this material, in addition to the use of MCH, allowed for a wetting/contact angle as low as 26°, greatly reducing nonspecific adsorption of proteins and improving device specificity. Furthermore, the aptasensor provided a large number of aptamer binding sites as a result of the abundance of hydroxyl groups in the polydopamine film and the Ag-S bonds supplied by the Ag NPs. This dense immobilization, combined with the hydrophilic properties of the polydopamine matrix and conductivity of Ag NPs, allowed the electrochemical aptasensor to achieve a LOD of 0.036 pM. Recovery varied between 94% and 108%, with an inter-device relative standard deviation (RSD) of 4.7%.

One-dimensional (1D) nanostructures such as nanowires and nanotubes allow increased detection sensitivity due to their unique electron transfer properties. These structures offer the potential of detecting a single molecule, as they avoid lateral current shunting and signal reduction that occur in two-dimensional (2D) materials [79]. As such, the use of nanowires and nanotubes is advantageous in label-free electrochemical aptasensing.

In a sandwich-type thrombin aptasensor designed by Zhang et al. [38], silver nanowires and nanoparticles (Ag NWs&NPs) were used to modify an ITO electrode. TBA1 was immobilized onto the nanowire modified electrode surface, while TBA2 was bound to ZnFe_2_O_4_ decorated with Pt nanoparticles (Pt NPs). The Ag NWs&NPs and Pt nanoparticles provided abundant binding sites for TBA1 and TBA2, respectively, and the addition of Ag NPs to the electrode surface reduced junction resistance in the Ag NWs by providing connections between nanowires. As for signal amplification, it was achieved through the synergistic catalytic activity of Ag NWs&NPs, ZnFe_2_O_4,_ and Pt nanoparticles towards H_2_O_2_, evidenced by increased conductivity when observed with amperometric and CV methods, with the subsequent addition of the aforementioned catalytic materials. Due to this synergy, along with the sandwich detection, higher sensitivity was achieved than in the aforementioned study that utilized Ag NPs only. The LOD obtained was as low as 0.016 pM. Furthermore, the aptasensor showed negligible responses to interfering substances and a recovery between 98.66–101.4%, reflecting the good specificity in real serum samples.

A more representative study in exploiting nanowires’ electrocatalytic activity for thrombin sensing was carried out by Zheng et al. [80] in which Pt-Pd nanowires were utilized in their aptasensor. As shown in Figure 6, signal amplification was achieved by utilizing a pseudo triple-enzyme cascade consisting of hemin/G-quadruplex, alcohol dehydrogenase (ADH), and the aforementioned Pt-Pd nanowires. This cascade sequentially produced H_2_O_2_, which was then catalytically reduced to H_2_O, thus mimicking the action of horseradish peroxidase (HRP) and eliminating the laborious labeling process. The Pt-Pd nanowires exhibited superior electrocatalytic activity as a result of the nano-branching structure which provided high surface roughness for molecular immobilization and low resistance for electron transfer. The electrode was a GCE, modified with Au nanoparticles to provide binding sites for TBA1. On the Pt-Pd nanowires, more TBA2 and ADH were immobilized which allowed for a sandwich configuration in the presence of thrombin. DPV measurements showed high sensitivity with a LOD of 0.067 pM and a wide linear range of 0.2 pM to 20 nM. Additionally, observable changes were triggered exclusively by thrombin, and the sensor exhibited a recovery between 95.84% and 106.7%.

Platinum nanomaterial was also used in the form of porous Pt nanotubes by Sun et al. [81], with a similar signal amplification strategy as the aforementioned aptasensor shown in Figure 6. The sensing electrode was a GCE modified with Au nanoparticles which allowed capturing of TBA1 onto the electrode surface. The porous nature of the Pt nanotubes provided a large surface area for the immobilization of TBA2, with the addition of hemin/G-quadruplex and glucose dehydrogenase (GDH) for signal amplification, enabling a sandwich assay detection of thrombin. Signal amplification was achieved through the synergistic effects of Pt, GDH, and hemin/G-quadruplex on the catalytic reduction of H_2_O_2_, achieving a LOD of 0.15 pM by DPV analysis. The aptasensor was also promising in terms of specificity, recovery, and reproducibility.

Two-dimensional metallic nanomaterials, particularly graphene and TMDC nanosheets, have become of particular interest in the field of biosensing due to a number of unique properties [82]. Graphene nanosheets, composed of a single layer of sp^2^ carbon atoms arranged into a hexagonal lattice [83], represented the first successful utilization of 2D nanomaterials. Their morphology allows for a very high aspect ratio and enhanced electron mobility, thereby sparking interest in the exploration of alternative nanomaterials of a similar structure. TMDCs have become a more attractive category of 2D materials than graphene due to their superior electrical properties [84]. Unlike graphene, these materials do not form a single layer, but rather a three-layered sandwich structure that is held together by strong covalent bonds. TMDCs can be modified to obtain a wide range of electrical properties. Hence, an interest in incorporating TMDCs into electrochemical biosensors has risen [85].

In a sandwich aptasensor designed by Wang et al. [54], a 2D TMDC material, namely tungsten selenide (WSe_2_), was used for enhancing the sensitivity of thrombin detection. WSe_2_ nanosheets formed a porous substrate platform with a large surface area. The metallic nature of Se and the unsaturated Se edges within the material offered superior electrical conductivity and electrocatalytic activity. In this biosensor, Au NPs were further used as a TBA2 carrier and modified by the signal probe biotin. Biotin allowed the capturing of streptavidin-conjugated alkaline phosphatase (SA-ALP), which was subsequently brought to the electrode surface once the sandwich assay had assembled in the presence of thrombin. SA-ALP, in turn, triggered electrochemical redox cycling for signal amplification with ALP hydrolyzing ascorbic acid 2-phosphate (AAP) to produce ascorbic acid (AA). This aptasensor achieved a LOD of 190 fg/mL and high stability, in addition to a recovery of 91.1% to 108.2%. Other 2D materials such as molybdenum disulfide (MoS_2_) have also been explored in the field of biosensing due to superior semi-conductive behavior, owing to high mobility and a direct intrinsic band gap [86]. In an aptasensor study reported by Lin et al. [87], MoS_2_ nanosheets were modified onto a Pt electrode, followed by immobilization of thrombin aptamers through thiol-mediated binding. Using the EIS method, a LOD of 267 fM was achieved with a linear thrombin concentration range of 2.67 pM to 267 pM.

Representative studies based on low dimensional metallic nanomaterials in thrombin sensing are summarized in Table 1. The enhanced aptasensors are arranged in order of dimensionality of utilized materials, starting with zero-dimensional nanoparticles up to 2D nanosheets. Characteristics pertaining to each design are summarized to highlight the complexity of design and performance of the various aptasensors. In general, metallic nanomaterials could significantly enhance the sensitivity of aptasensors to the fM-pM levels due to their excellent electrical and electrochemical properties. Increasing morphological complexity of nanomaterials did not necessarily lead to higher sensitivity but increased the linear responsive range instead. The simplest aptasensor, utilizing only Au NPs, achieved one of the lowest LODs around 0.14 fM, while Ag NW based sensors offered linear ranges of around five orders of magnitude. Two-dimensional materials for biosensing are in their early stages and still have room to improve in terms of sensitivity and linear range.

### 4.2. Porous Nanomaterial-Based Thrombin Electrochemical Aptasensors

Hollow and porous nanomaterials were utilized in biosensing due to their enhanced surface area and volume provided [88], allowing for a higher aptamer loading capacity [89]. Metal oxide nanospheres, metal organic frameworks (MOFs), and nanochannels are examples of such materials incorporated into electrochemical aptasensor designs. MnO_2_ nanospheres are an example of metal oxide nanospheres employed by Shuai et al. [90]. In this study, thiolated TBA1 was immobilized onto a GCE modified with N-doped graphene oxide (N-GO) and Au NPs, while TBA2 was linked to an Au NP-MnO_2_ nanohybrid. The MnO_2_ nanospheres displayed a highly porous structure with a large surface area as a result of the hierarchical nanosheet composition, allowing for the immobilization of significant amounts of Au NPs. Signal amplification was realized through hybridization chain reaction (HCR) which was achieved by attaching HRP to the electrode through a streptavidin-biotin reaction. By combining the superior surface properties of the N-GO, enhanced electrical conductivity of Au NPS, increased surface area of MnO_2_ nanospheres, as well as signal amplification by HRP, an ultrasensitive aptasensor was obtained with a LOD as low as 27 aM and a wide linear range of detection extending between 0.1 fM and 0.1 nM. Furthermore, the characteristics of the aforementioned materials imparted excellent stability and reproducibility (RSD = 4.2%) of the aptasensor and a recovery between 91.0% and 106%. Another kind of nanosphere utilized in electrochemical thrombin aptasensors is Cu_2_O nanospheres. In a study by Zhang et al. [91], the strong affinity between Cu_2_O and ssDNA was exploited to synthesize Cu_2_O@aptamer nanospheres, where aptamers were evenly distributed throughout and on the surface of the hollow Cu_2_O spheres. These nanospheres were modified onto a gold electrode and, due to their porous nature, thrombin was able to diffuse within the structure, binding to the aptamers present in the hollow interior. This setup allowed for a greater amount of thrombin binding and a LOD of 0.33 pM was achieved. However, the selectivity of this aptasensor was slightly compromised in the case of BSA proteins due to the high affinity of BSA to the Cu_2_O nanospheres.

Additional porous nanomaterials used in electrochemical aptasensing are MOFs. MOFs are a type of crystalline materials composed of metal ions held together by organic linkers and are characterized by their large surface area, flexible porosity, and large loading capacity, allowing for their incorporation in biosensing [92]. In a study conducted by Yang et al. [28], MOF composed of Co^2+^ and 2-amino terephthalic acid (NH_2_-H_2_BDC) was used to load Pt nanoparticles within and on the surface of the structure, thereby producing PtNPs@Co(II)MOFs@PtNPs composite for thrombin detection. The aptasensor utilized a sandwich design, with a GCE modified with Au nanoparticles carrying TBA1 and the MOF-based composite immobilizing TBA2. Increased loading of Pt NPs—TBA2 bioconjugate revealed exceptional signal amplification, with Co(II) to Co(III) electron transfer mediating the electrochemical signal. The enhanced electron transfer, aptamer binding sites, and electrocatalytic activity of PtNPs@Co(II)MOFs@PtNPs achieved a LOD as low as 33 fM. Moreover, the sensor exhibited high stability, reproducibility, and real sample recovery between 91.32% and 105.39%. Similarly, Xie et al. [93] incorporated Fe-MIL-88 MOFs with hemin (hemin@MOFs) in a sandwich aptasensor design to increase the catalytic lifetime of the inherently unstable hemin. Au nanoparticles were used to modify both hemin@MOFs and the GCE, onto which TBA1 was immobilized. Au/hemin@MOFs are further conjugated with glucose oxidase to serve as the platform for TBA2, in addition to acting as the redox mediator and catalyst. The dual catalytic scheme, provided by glucose oxidase and hemin@MOFs, allowed for dramatic signal amplification and a detection limit of 0.067 pM was achieved. The aptasensor showed similar results in comparison with conventional ELISA testing in real serum samples, indicating the practical potential of this aptasensor for thrombin detection in complex samples.

Ni MOF was also used in a sandwich aptasensor designed by Wu. Et al. [94]. A GCE modified with Au NP film was used to immobilize TBA1, whereas the Ni MOF was used to construct the TBA2 signal probe. The Ni MOFs were integrated with 4,4′,4” -Tricarboxytriphenyl- amine (H_3_TCA) ligands, which are redox-active, and the structure was stabilized with magnetic nickel cluster node crosslinking to prevent the loss of electrochemical activity. The Ni MOF structure was further modified with Au NPs, to which binding of TBA2 occurred. Due to the wrinkle ball-flower structure of Ni MOF, an extended surface area for Au NP binding was provided, thereby improving electron transport. DPV measurements showed a linear decrease in current output over the range of 0.05 pM to 50 nM of thrombin concentration, with a LOD of 0.016 pM. Moreover, the aptasensor was highly selective and stable, as it was unresponsive to interferents even with 10-fold higher concentrations, and retained 98.2% of its initial current after 19 days of storage.

In addition to porous nanomaterials directly used for electrode materials, 3D nanostructures could be utilized in other ways to enable thrombin detection. Specifically, nanochannels could be used as separators between two chambers filled with an electrolytic solution to allow the selective transport of molecules, reflected as changes in conductance [95]. For example, TBA entrapped within Cytolysin A biological nanopores could display the conformational change in TBA upon binding to thrombin [96]. In a typical study in using nanochannels for thrombin detection [97], a porous anodic alumina (PAA) membrane was utilized. This PAA membrane contained a high density of nanochannels, allowing for greater signal amplification. Two half-cells with KCl solution were used rather than the typical three-electrode setup, with Ag/AgCl electrodes as the anode and cathode, and the PAA membrane was placed between the half-cells. TBA was immobilized onto the inner walls of the nanochannels. As shown in Figure 7, the principle of this aptasensor worked on the change in space occurring within the nanochannels upon binding of TBA to thrombin, converting from a flexible structure to a rigid G-quadruplex, creating a steric hindrance for the transport of ions across the nanochannels. As thrombin concentration increased, a decrease in ionic current was observed and a LOD of 1 pM was obtained. The biosensor displayed excellent selectivity, however, the performance started to decline after four days of bioactivity testing.

The same concept was slightly modified and employed in a three-electrode aptasensing system by Li et al. [98]. In this study, a polyester terephthalate (PET) membrane coated with Au NPs and embedded with multiple ion channels acted as the electrode, and amine-modified aptamers were immobilized on the inner surface of the ion channels by binding to PET carboxyl groups. The aptasensor followed a “signal-on” mechanism, and the signal was generated through [Ru(NH_3_)_6_]^3+^ ions which were bound to the negative backbone of the aptamer in the absence of thrombin. Upon aptamer-thrombin binding, these cations were displaced and pushed towards the electrode surface through the channels due to the positive nature of thrombin, consequently, a current response was generated and increased with thrombin concentration. Although very high concentrations of thrombin tended to decrease the signal as a result of crowding in the inner diameter of the ion channels that restrict the ion flow, this aptasensor achieved a low LOD at 0.6 pM. Moreover, this aptasensor exhibited high regeneration and signal retention at 99.3%, as cations and thrombin could be easily pushed out of the ion channels based on concentration and potential differences.

Studies based on these porous nanomaterials in thrombin sensing are summarized in Table 2. When integrated with metallic nanoparticles, hierarchical porous composites allow the combination and optimization of electron transfer, aptamer binding, and electrocatalytic activities at the same time, extending the LOD to the attomolar level and the linear range to six orders. Aptasensors based on nanochannels differed significantly in terms of design in contrast to other metallic nanomaterial-based aptasensors, and this difference was reflected in sensors with a relatively poor sensitivity but simple fabrication and easy regeneration in comparison to other sensors.

### 4.3. Carbon Nanomaterial-Based Thrombin Electrochemical Aptasensors

While metallic or metal-related nanomaterials have obvious advantages in terms of electrocatalytic properties, carbon nanomaterials are favorable largely due to the presence of appropriate functional groups for simplifying aptamer immobilization [83]. Furthermore, these functional groups impart hydrophilicity that allows the incorporation of inorganic materials to form nanocomposites with enhanced performance for biosensing [99].

Graphene, a previously addressed 2D carbon nanomaterial, has been used in biosensor fabrication owing to its advantageous features such as large surface area, enhanced electrical activity, and availability of active sites [99,100]. Graphene Oxide (GO) is the oxidized form of graphene, in which the 2D structure is retained, though electrical conductivity is greatly reduced [82,101]. Conductivity can be restored by reduction to produce reduced graphene oxide (rGO) which is preferred in aptasensors over pure graphene due to its hydrophilic nature and increased availability of functional groups for immobilization of aptamers [102].

Ahour et al. [103] designed a pencil graphite electrode (PGE) modified by rGO for thrombin detection by utilizing the intrinsic catalytic ability of the thrombin aptamer sequence, originating from guanine base oxidation. Modification of PGE with rGO revealed an increased DPV peak, owing to the larger surface area provided by rGO and the superior electrical conductivity. The working principle relied on the desorption of the oxidized signal-generating aptamers from the rGO modified electrode surface in the presence of thrombin as a result of the binding-induced conformational changes. This event then led to decreased guanine oxidation, which translated as a decreased DPV signal. The LOD achieved was 0.07 nM, and the current was reduced to 92% of its original value after 1 week of storage. The recovery obtained in serum samples ranged from 98% to 103%.

In addition to TBA immobilization, graphene-based materials can also be used to immobilize other nanostructures to form hybrid nanocomposites. Due to the presence of surface defects and reactive functional groups, which offer abundant anchoring sites for material loading in case of direct mixing and provide the appropriate surface environment for the nucleation and growth of inorganic nanoparticles in case of in situ synthesis [99], well-prepared composites allow for synergistic electrocatalytic activity and enhanced sensitivity of the aptasensor [69]. In a study by Zhang et al. [104], Fe_2_O_3_@graphene cast GCE was incorporated into an electrochemical aptasensor for the detection of thrombin, where Fe_2_O_3_ was added to graphene through atomic layer deposition (ALD). Both CV and EIS measurements indicated improved conductivity as a result of the increased surface area of the electrode upon sequential modification with graphene and Fe_2_O_3_ NPs. In addition to the advantages of graphene, the Fe_2_O_3_ NPs have an intrinsic electrochemical activity that can work to enhance the sensitivity of the biosensor. Using the DPV technique, it was found that the redox current of Fe(CN)_6_^3−/4−^ was inversely proportional to the concentration of thrombin, achieving a LOD of 1.0 pM and a linear detection range of 10 pM to 4.0 nM.

Another graphene-based nanocomposite utilized in an aptasensor by Zhang et al. [69] was graphene-porphyrin (GN-Por), modified onto a GCE. GN-Por combined the advantageous features of graphene with the excellent electrochemical activity of porphyrin, and both materials allowed thrombin aptamer immobilization through π-π stacking. The redox current of Fe(CN)_6_^3−/4−^ generated by porphyrin decreased linearly with the increase in thrombin concentration at a range of 5 nM to 1.5 μM, due to electron transfer hindrance. The LOD obtained was 0.2 nM, with a recovery range of 98% to 110% and excellent stability.

Qin et al. [24] utilized Ag NP-decorated rGO in an electrochemical aptasensor and incorporated the resulting nanocomposite with a gold UME. The UME allowed for enhanced sensitivity and lower background noise of the biosensor, in comparison to a macroelectrode. The aptasensor in this study exploited the double binding sites of the thrombin analyte by setting up a sandwich assay detection. Apt29 (TBA2) was immobilized onto the Au microelectrode, while Apt15 (TBA1) self-assembled onto the Ag NPs-modified rGO (TBA1-AgNP-GO) which acted as the signal probe. Square wave voltammetry (SWV) was used to monitor oxidation current in response to thrombin concentration. The greater the concentration of the thrombin, the more TBA1-AgNP-GO assembled onto the electrode surface, allowing for increased oxidation of Ag NPs and a stronger current. A detection limit of 0.03 nM was obtained, indicating good sensitivity of the biosensor, in addition to the high selectivity of the sensor in response to interference proteins.

Another type of carbon nanomaterial, CNTs, both single-walled carbon nanotubes (SWCNTs) and multi-walled carbon nanotubes (MWCNTs) [105], have been incorporated in electrochemical biosensing due to their exceptional electrical and electrochemical properties, allowing for higher sensitivity and lower LOD owing to their enhanced conductivity in comparison to other carbon-based materials [106,107]. Immobilization of biosensing molecules is highly convenient due to the large surface area [108] and the ability of CNTs to bind directly to aptamers without resorting to chemical modifications [23], similar to graphene. The main challenge faced when utilizing CNTs in biosensing is their inherent hydrophobic nature and high surface energy, making them difficult to handle in a controlled manner. However, such issues can be resolved through modification of CNTs by covalent and noncovalent surface functionalization, allowing for easier dispersion [109,110].

A simple aptasensor was demonstrated by Park et al. [23] by direct immobilization of Apt29 on an SWCNT modified GCE. The working principle of this aptasensor was based on the different binding affinities of the TBA to the SWCNTs and the thrombin. The higher the concentration of thrombin, the greater the number of immobilized TBAs removed from the SWCNT electrode due to the competitive binding to thrombin, which was indicated by the current increase measured by CV. However, this principle led to a high LOD (10 nM), owing to the fact that TBA always has a considerable affinity to SWCNTs, therefore, high concentrations of thrombin were required to remove TBAs from the electrode surface. Nevertheless, this approach enabled a simple and less expensive detection of thrombin by exploiting the intrinsic catalytic nature of the aptamer guanine bases, enhanced by the Ru(bpy)^2+^ mediators.

In order to achieve a lower LOD, a more complex aptamer/electrode was designed and studied to directly measure the binding activity of thrombin on aptamer-coated electrodes [111]. MWCNTs were incorporated into a nanocomposite comprising TiO_2_ nanocrystals, chitosan, and a synthetic Schiff base (TiO_2_-MWCNT/CHIT-SB), which was then used to modify the GCE surface. The primary advantage of this sensor was the enhanced surface area provided for aptamer immobilization, as each of the four components of the nanocomposite can interact with DNA, therefore working synergistically to allow for increased adsorption of the thrombin aptamer to the surface of the electrode. Furthermore, characterization of the electrochemical properties through EIS, CV, and DPV showed the excellent electron transfer properties of TiO_2_-MWCNT that partially compensated for the poor conductivity of CHIT-SB. The electrode modified by TiO_2_-MWCNT/CHIT-SB composite produced greater sensitivity than that modified with either TiO_2_ or TiO_2_-MWCNT. The aptasensor allowed for a LOD as low as 1.0 fM, high selectivity and stability, with 94% of the initial signal retained after one month.

An even lower LOD of 0.1 fM was achieved by utilizing a CNT/ZnCr-LDH hybrid (Figure 8a) modified gold electrode [112]. Layered double hydroxides (LDH) allow the detection of analytes in lower concentrations due to their ability to pre-concentrate the analyte on the surface of the electrode to a small volume. This material, however, suffers from poor electrical conductivity, which was compensated in this design by the advantageous electrical properties of SWCNTs. Though aptamers can directly bind to LDH through the bases and phosphate groups of their sequences, 5′-NH_2_ modifications of aptamers allow them a vertical orientation when attaching to LDH surfaces, enabling a denser immobilization. Thrombin concentration was measured using DPV, where an increase in thrombin concentration caused a decrease in current, due to the insulating properties of the aptamer-thrombin complex, as illustrated in Figure 8b. The aptasensor obtained a linear range of 5 fM to 12 nM and a LOD of 0.1 fM (Figure 8c), which were the widest range and lowest LOD achieved up to that point in time. Additionally, this sensor displayed excellent selectivity and a 94% retention of the initial signal after 30 days of storage.

Similarly, a nanocomposite incorporating MWCNTs was utilized in a study by Jamei et al. [45] to modify a screen-printed carbon electrode (SPCE) for the detection of thrombin. In addition to MWCNTs, a carbon nanomaterial known as fullerene (C60) was also used, which served to increase the surface area of the nanotubes. MWCNTs were further modified with polyethylenimine (PEI), which provided an abundance of amine groups for aptamer immobilization. Polymer quantum dots (PQdot) were also added to the nanocomposite through electrostatic interactions. PQdots are formed of a carbon core with attached polymer chains and have semiconductor characteristics. These nanomaterials were combined to form a C60/MWCNTs-PEI/PQdot nanocomposite. Although the PQdots decreased conductivity as a result of their semiconductor nature, they worked synergistically with the remaining nanocomposite components to increase surface area and aptamer immobilization sites, allowing for a sensitive aptasensor with a low LOD of 6 fM. Furthermore, the aptasensor was highly reproducible, with 10 sensors manufactured under the same conditions producing a relative standard deviation of 3.6%.

Another carbon nanomaterial used as an electrochemical aptasensor for thrombin detection is carbon nanocages, which are three-dimensional structures composed of a graphitic shell. The large surface area of these structures and availability of reactive groups allow for dense immobilization of aptamers. A study by Gao et al. [113] deployed carbon nanocages (CNCs) in an aptasensor, where a sandwich assay was used, and the signal probe was the Pt nanoparticle modified carbon nanocage with attached TBA (TBA–Pt NPs/CNCs). A gold electrode was used, on which thrombin aptamers were immobilized. DPV measurements were used to track thrombin concentration changes, with a current decrease observed with increased concentration due to the increasing electron transfer hindrance by the inert thrombin. The result was a highly sensitive, sandwich-type aptasensor with a LOD of 0.01 pM. The sensitivity of the device was attributed to the large surface area of the CNCs, allowing for ample immobilization of Pt NPs that significantly amplified the electrochemical signal through the catalytic reduction of H_2_O_2_. However, the stability of the sensor was lower than previously discussed aptasensors, with the signal dropping to 86.1% of its original value following only 14 days of storage.

Some key parameters of carbon nanomaterial-based thrombin aptasensors are summarized in Table 3. On average, these electrochemical aptasensors were inferior to metal-enhanced sensors in terms of sensitivity. The main advantage of carbon nanomaterials is their capability of direct binding to aptamers, easing the TBA functionalization process. In addition, the π-π interaction and rich functional groups in carbon nanomaterials allow them the ability to anchor a variety of other nanomaterials, offering the opportunity and freedom to maximize the synergistic effects between material components and thus boost sensors’ electrocatalytic activity. As a result, the optimized sensors showed both superior LOD (fM level) and a wide linear range (6–7 orders).

### 4.4. Magnetic Nanoparticle (MNP)-Enhanced Thrombin Electrochemical Aptasensors

MNPs have been introduced into the field of biosensing, as they offer an additional degree of controllability to increase sensitivity. In addition to the large surface area and enhanced electrical conductivity of MNPs that can be used for electrode surface modification, super magnetism in MNPs allows a fast response in the presence of a magnetic field and transport of redox-active species to the electrode surface, resulting in a rapid and sensitive analysis [114].

The magnetic properties of MNPs were demonstrated in a study by Zhang et al. [115] via a homogeneous sandwich electrochemical aptasensor. As shown in Figure 9, TBA1 was modified onto the MNPs or magnetic beads (MBs), while TBA2 was immobilized on hydroxyapatite (HAP) nanoparticles. In the presence of thrombin, MNP-TBA1 and HAP-TBA2 sandwich structures formed which could then be collected via a magnetic field, separating out the unconjugated NPs. These sandwich conjugates were then transferred to the electrode surface, where HAP generated a cascade reaction leading to the formation of molybdophosphate precipitate on the electrode surface. The output of this study was a highly sensitive aptasensor, detecting thrombin at concentrations as low as 0.03 fM with a linear range of 0.1 fM to 1.0 nM. This homogeneous design was compared to a heterogeneous version where aptamers were immobilized directly onto the electrode surface. The homogeneous one displayed greater sensitivity and precision, due to the enhanced electron transfer, and minimal steric hindrance brought about by avoiding the presence of large biomolecules on the electrode surface.

Magnetic nanoparticles also facilitate the self-assembly of nanostructures under the direction of a magnetic field, such as Fe_3_O_4_@Au nanocomposite in the study conducted by Zhu et al. [116]. This composite combined the magnetic property of Fe_3_O_4_ NPs with the aptamer anchoring property of Au NPs. Double-stranded DNA composed of thrombin aptamer and its complementary strand were immobilized onto the Fe_3_O_4_@Au NPs to form a nanocomposite, which then self-assembled onto a magnetic GCE. Upon exposure to thrombin, competitive binding of the aptamer strand to thrombin led to its dissociation from its complementary strand, exposing a binding site for Pb^2+^-dependent DNAzyme on the complementary strand. This DNAzyme could then cleave it from its attachment to the magnetic nanocomposite. Only double-stranded DNA remained on the electrode surface, and the concentration of thrombin was measured through the change in signal output by methylene blue, which was physically adsorbed onto the thrombin aptamer. The higher the concentration of thrombin, the more aptamer dissociation and the less methylene blue present at the electrode surface. A LOD of 1.8 pM and a broad linear range from 5 pM to 5 nM were obtained with DPV. Moreover, the aptasensor was tested on serum samples of pregnant women with recovery between 93.67% and 104.21% and RSD lower than 1.28%. This is significant as physiological changes occurring with pregnancy lead to an increased thrombotic state, necessitating the monitoring of coagulation factors in this patient group [117].

### 4.5. Polymer-Based Thrombin Electrochemical Aptasensors

As aforementioned, detection of thrombin is difficult in whole blood samples due to the biofouling effect, and instead, it has traditionally been done through isolated serum samples [63]. Introducing polymeric materials into electrochemical aptasensors can prevent such nonspecific adsorption of proteins [118]. Hydrophilic polymeric materials achieve antifouling by the formation of a hydration layer through binding to interfacial water molecules [119,120], and as such, provide a method for achieving thrombin detection through whole blood analysis [63]. Polymers are traditionally known as insulating materials. However, a class of polymers known as conducting polymers is characterized by a π-electron backbone, allowing for electrical conductivity [121]. These materials can be used to modify the electrode surface and impart antifouling properties while simultaneously improving conductivity [71].

In an aforementioned study by Xu et al. [78], a self-polymerizing dopamine matrix was introduced onto the surface of the working electrode, which exhibited hydrophilicity, reducing the biofouling of the electrode. In an earlier study by Sun et al. [63], carboxymethyl-PEG-carboxymethyl (CM-PEG-CM) was deposited on the surface of a GCE as a hydrophilic polymeric coating. In this study, however, the aim was to detect thrombin in whole blood samples, rather than a limited detection in blood serum. Great biocompatibility and antibiofouling effect were observed due to the highly hydrophilic nature of CM-PEG-CM. The contact angle of the surface was found to be 11.9° due to the carboxylate groups found in the coating polymer. The study not only succeeded in detecting thrombin in whole blood samples but successfully detected thrombin at concentrations as low as 15.6 fM, indicating the excellent anti-biofouling design of the aptasensor.

Detection of thrombin in whole blood samples was also studied by Sun et al. [22] using an ITO electrode modified by aliphatic hyperbranched polyester microspheres with carboxylic acid functional groups (HBPE-CA) with the structure depicted in Figure 10a. Due to the presence of these functional groups on the microspheres, HBPE-CA was highly hemocompatible, as its presence led to increased coagulation test times and did not lead to platelet activation. TBA was then immobilized on the modified ITO surface, and thrombin concentration was measured through DPV. With increasing thrombin concentration, electron transfer was further impeded between the electrode and the redox probe [Fe(CN)_6_]^3−/4−^, translating as a decreased current output as illustrated in Figure 10b. A lower LOD was achieved at 0.90 fM, with a linear detection range of 10 fM–100 nM in serum samples, with similar results obtained in whole blood samples. Moreover, the sensor showed good specificity in response to other serum proteins and retained 92.8% of its signal after 20 days.

In a more recent study by Niu et al. [122], hyperbranched polyester was once again used as an anti-biofouling coating, this time in the form of heparin-mimicking hyperbranched polyester nanoparticles (HBPE-SO_3_ NPs). HBPE-SO_3_ NPs and positively charged Au NPs were used to modify a GCE and exhibited anticoagulant activity. Similar to heparin, the anticoagulant properties can be attributed to the presence of sulfonate groups in the HBPE-SO_3_ NPs. The modified electrode resulted in a significant decrease in biofouling, inhibiting platelet adhesion and activation, and less than 5% hemolysis, all indicating the good hemocompatibility of the modified electrode. As for the principle of thrombin detection, this aptasensor utilized the steric hindrance occurring with conformational changes in TBA upon binding to thrombin, which would subsequently impair electron transfer, thus increasing resistance. In comparison to the previous study by Sun et al. which also utilized hyperbranched polyester [22], the aptasensor in this study could also detect in whole blood analysis, though was less sensitive, achieving a LOD of 0.031 pM and a linear range of 2.70 pM to 270 nM [122].

The polymers utilized in the aforementioned studies were not conductive, though they showed effective antifouling. Conducting polymer, namely a conductive supramolecular polymer hydrogel (CSPH), was introduced in a sandwich electrochemical aptasensor by Wang et al. [71]. The CSPH in this case was made with an aniline (AN) 3-aminophe- nylboronic (ABA) and polyvinyl alcohol (PVA) solution, forming a conductive porous network. The CSPH also exhibited high hydrophilicity, with the CSPH- modified GCE demonstrating a water contact angle of 31.7°. TBA1 was immobilized onto the CSPH modified GCE electrode surface, while TBA2 was immobilized onto magnetic nanoparticles (MNP-TBA2) to act as the signaling probe. In the presence of thrombin, a sandwich assay formed, allowing for DPV measurements of thrombin concentrations. CSPH combined both antifouling properties and enhanced conductivity to improve the performance of the aptasensor, eliminating the need for additional materials to mediate electron transfer and block unmodified immobilization sites. The aptasensor demonstrated good specificity, antifouling performance, and recoveries ranging between 95.2% and 106.3%. A detection limit of 0.64 pM was achieved, however, analysis was done in serum samples rather than whole blood analysis.

Key results of polymer-based thrombin aptasensors are summarized in Table 4. On average, the performance of these aptasensors was superior to those in the carbon-based category and comparable to the metal-enhanced sensors. However, they did not achieve a LOD as low as the best performing aptasensor in the aforementioned category. The main distinctive properties of polymer-based aptasensors are their biocompatibility and anti-biofouling properties, with various aptasensors detecting thrombin in whole blood analysis, eliminating the need for serum isolation.

## 5. Conclusions

The detection of thrombin is essential for the early diagnosis of various diseases. Label-free techniques, particularly electrochemical aptasensors, offer simplicity, specificity, sensitivity, and speed in detection, though challenges in its employment in a clinical setting still prevail. This review illustrated the different materials that have been utilized in label-free electrochemical aptasensors for the detection of thrombin in an attempt to overcome shortcomings such as higher than expected detection limits, biofouling, and restrictions on use in whole blood samples.

Various nanomaterials were utilized, each with distinctive properties aiding in solving specific challenges in biosensing. Particularly, metallic nanomaterials have excellent charge transfer properties and are easy to bind with TBAs via self-assembling, and thus could significantly enhance the sensitivity of the aptasensors, especially in sandwich assay detection. Carbon nanomaterials have strong π-π interaction and rich function groups that allow them to bind TBA and other nanomaterials directly, easing the fabrication process and providing freedom for sensor design. Magnetic particles offer an additional degree of controllability to allow a fast response in the presence of a magnetic field and transport of redox-active species to the electrode surface for rapid and sensitive analysis. Polymer material based aptasensors, meanwhile, distinguish themselves by their biocompatibility and anti-biofouling properties, allowing them to detect thrombin in whole blood analysis.

Considering the fact that each material component cannot provide all required functions and the best performance was obtained from those aptasensors with an integration of multi-material components, future developments should work on combining functional nanomaterials to maximize the synergistic effects between them and thus boost sensors’ electrocatalytic activity and overall performance. Carbon nanomaterials and porous nanomaterials could be the best candidates for anchoring other material components for this purpose. At the same time, detection of thrombin in whole blood is equally important for easy and fast diagnosis.

## Figures and Tables

**Figure 1 biosensors-12-00253-f001:**
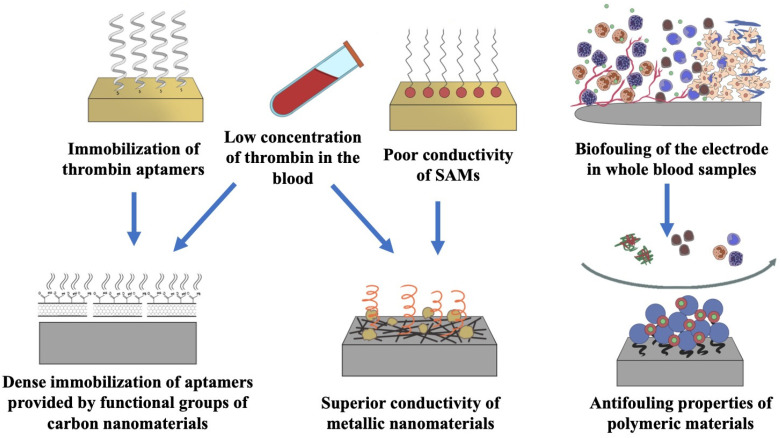
Immobilization of aptamers, low concentration of thrombin in the blood, poor conductivity of SAMs, and biofouling of electrodes in whole blood are the main challenges in the electrochemical detection of thrombin. These issues and problems could be resolved by utilizing nanomaterials and nanostructures.

**Figure 2 biosensors-12-00253-f002:**
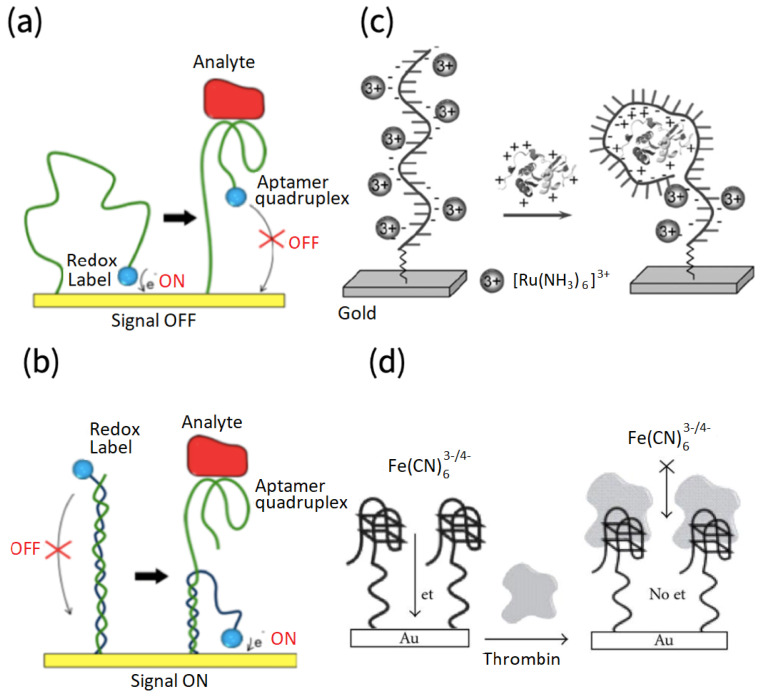
Conformational changes in aptamer structure due to binding to the analyte. When the electrochemically active probe is attached to the aptamer, binding-induced conformational changes bring the probe either farther (**a**) or closer (**b**) to the electrode surface, resulting in signal off or signal on, respectively. Adapted from [39]. In (**c**), binding of the analyte to aptamer reduces the electrostatic interaction between [Ru(NH_3_)_6_]^3+^ and aptamer, which results in a decrease in the reduction peak. Adapted from [40]. In (**d**), the binding of the analyte to the aptamer hinders the access of the solution-based redox probe to the electrode surface. Adapted from [41].

**Figure 3 biosensors-12-00253-f003:**
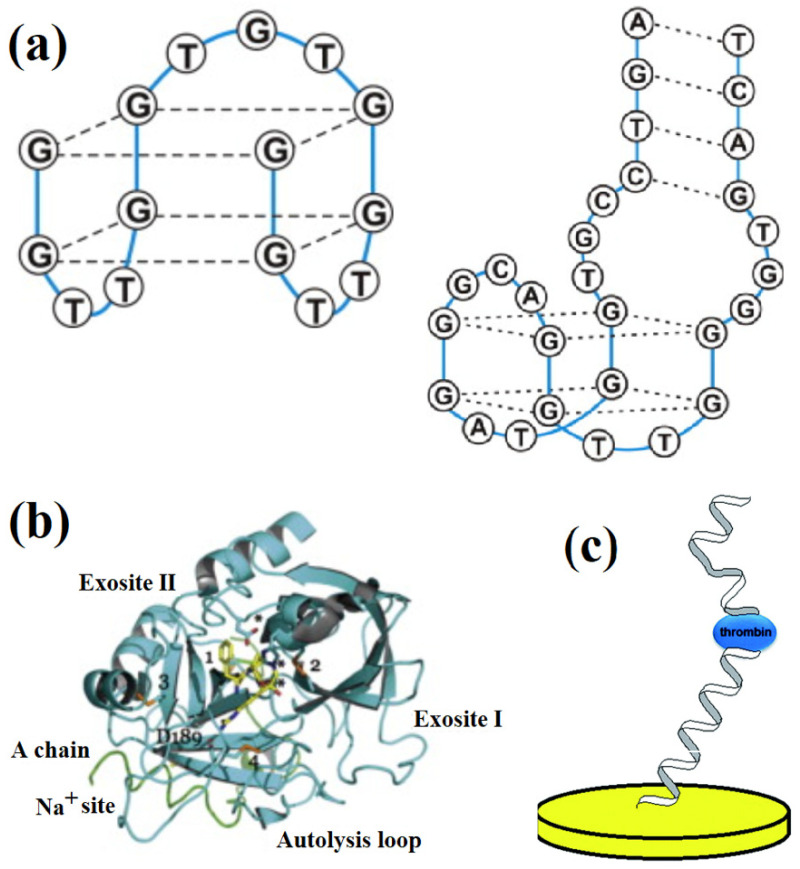
(**a**) G-quadruplex structure of Apt15 and Apt29, and (**b**) binding sites of thrombin. Adapted from [3]. (**c**) Sandwich assay setup utilizing both thrombin aptamers. Adapted from [61].

**Figure 4 biosensors-12-00253-f004:**
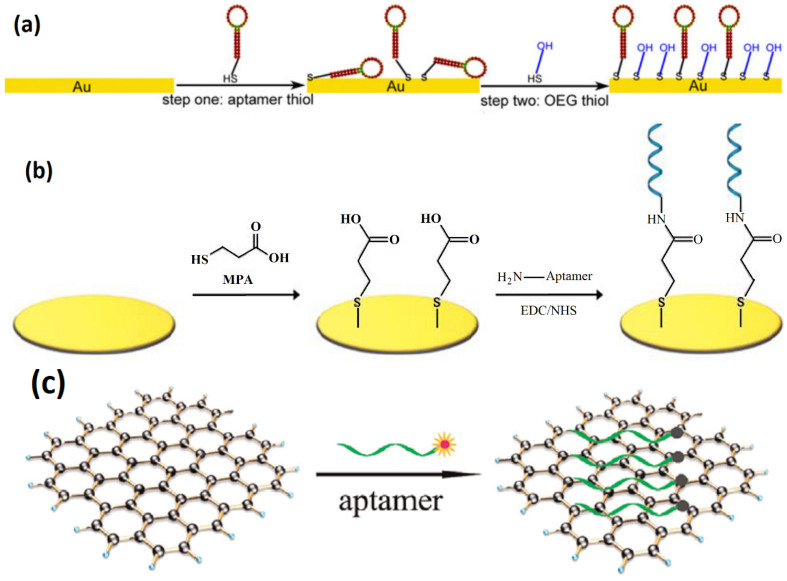
Aptamer immobilization via Au-S bonds (**a**), adapted from [68]; covalent attachment (**b**), adapted from [33]; and direct π-π stacking (**c**), adapted from [70], respectively. Incubation with OEG thiol in (**a**) blocks available functional groups after aptamer immobilization, thereby reducing nonspecific protein adsorption.

**Figure 5 biosensors-12-00253-f005:**
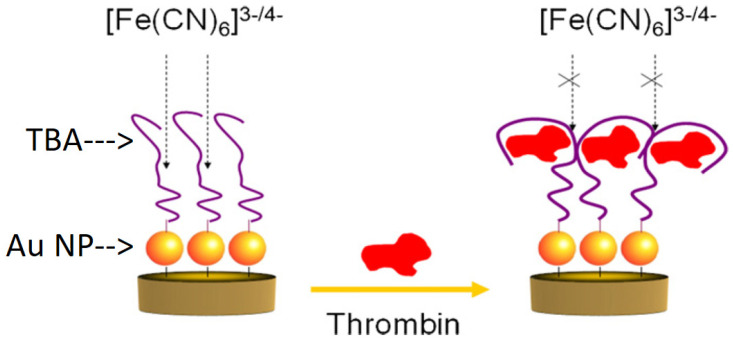
Au NP enabled thrombin biosensors. Conformational changes of TBA (immobilized on Au NPs) upon binding to thrombin vary the charge transfer resistances in the presence of [Fe(CN)_6_]^3−/4−^ redox couple. Adapted from [76].

**Figure 6 biosensors-12-00253-f006:**
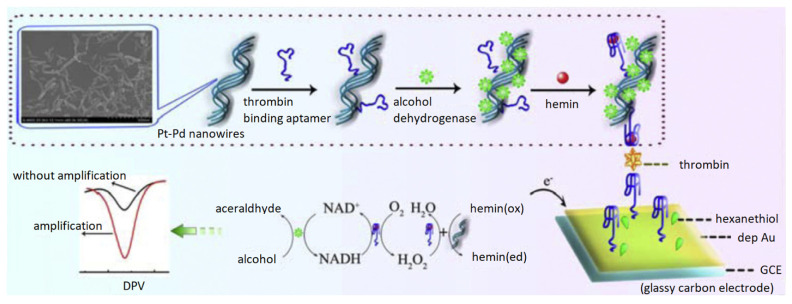
Pt-Pd nanowire-based pseudo triple-enzyme electrochemical aptasensor for thrombin detection. Adapted from [80].

**Figure 7 biosensors-12-00253-f007:**
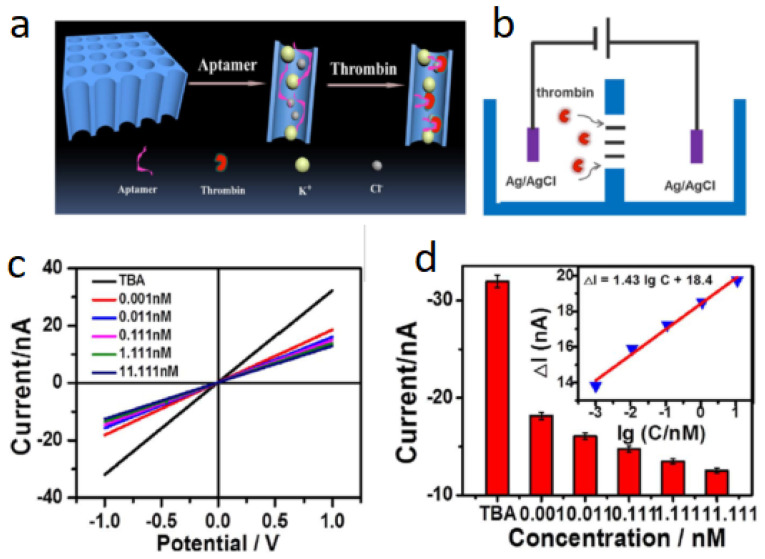
Nanochannel-based electrochemical aptasensor. (**a**) Thrombin immobilized within nanochannels, (**b**) counter-counter design of the aptasensor, (**c**) the I–V profiles obtained with different concentrations of thrombin, and (**d**) the ionic current versus thrombin concentration. Adapted from [97].

**Figure 8 biosensors-12-00253-f008:**
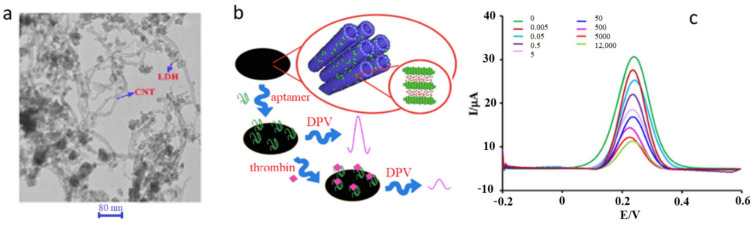
CNT composite-based thrombin aptasensor. (**a**) TEM images of CNT/ZnCr-LDH nanohybrid, (**b**) sensing mechanism of the aptasensor, and (**c**) DPV signals of aptasensor for different thrombin concentrations. Adapted from [112].

**Figure 9 biosensors-12-00253-f009:**
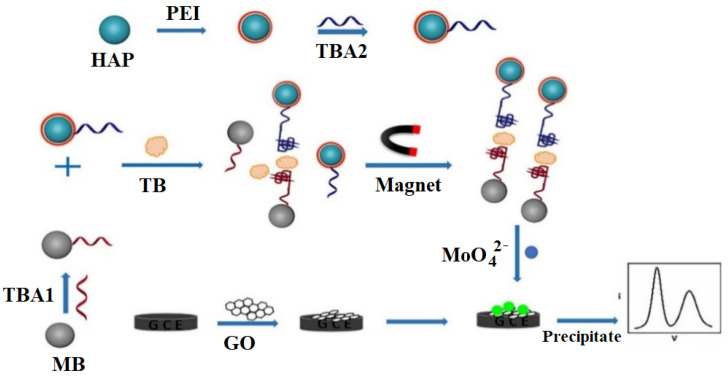
Principle of the magnetic nanoparticle-based aptasensor developed by Zhang et al. (MB: magnetic bead). Adapted from [115].

**Figure 10 biosensors-12-00253-f010:**
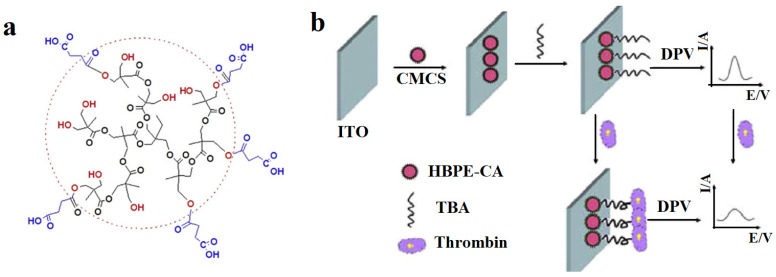
Aptasensor designed for thrombin detection in whole blood samples. (**a**) The chemical structure of HBPE-CA microspheres and (**b**) fabrication and principle of aptasensor. Adapted from [22].

**Table 1 biosensors-12-00253-t001:** Summary and comparison of low-dimensional metallic nanomaterials enhanced aptasensors.

Material Category	Detailed Electrode Material	Aptamer Sequence	Analytical Method	LOD (fM)	Linear Range	Others	Reference
Metallic nanoparticle (0D)	Au NPs (on Au electrode)	TBA1: 5′-SH-(CH_2_)_6_-GGT TGG TGT GGT TGG-3′ TBA2: 5′-SH-AGT CCG TGG TAG GGC AGG TTG GGG TGA CT-3′	DPV	0.1429	1 fM to 6 pM	Directly bound TBA	[77]
	Ag NPs @ dopamine (on GCE)	5′-SH-(CH_2_)_6_-GGT TGG TGT GGT TGG-3′	EIS	36	0.1 pM to 5.0 nM	Conductive and hydrophilic	[78]
Metallic nanowire/tube(1D)	Ag NWs&NPs/ZnFe_2_O_4_ NPs (on ITO)	Apt1: 5′-NH_2_-(CH_2_)_6_-GGT TGG TGT GGT TGG-3′ Apt2: 5′-SH—(CH_2_)_6—_AGT CCGTGG TAG GGC AGG TTG GGG TGA CT-3′	Amperometric I-t	16	0.05 pM to 35 nM	Sandwich assay design	[38]
	Pt-Pd NWs (on GCE)	5′-SH-(CH_2_)_6_-GGT TGG TGT GGT TGG-3′	DPV	67	0.2 pM to 20 nM	Triple enzyme cascade	[80]
	Pt Nanotubes (on GCE)	5′-SH–(CH_2_)_6_–GGT TGG TGT GGT TGG-3′	DPV	150	0.4 pM to 30 nM	Sandwich assay design	[81]
Metallic nanosheet (2D)	Au NPs/WSe_2_ (on GCE)	TBA1: 5′-biotin-TEG linker-GGT TGG TGT GGT TGG-3′ TBA2: 5′-NH_2_-TEG linker-AGT CCG TGG TAG GGC AGG TTG GGG TGA CT-3′	DPV	190	0–1 ng/mL	Sandwich assay design	[54]
	MoS_2_ (on Pt)	TBA (12T): 5′-(Thiol-C6) TTT TTT TTT TTT GGT TGG TGT GGT TGG-3′	EIS	267	2.67 pM to 267 pM	TMDC semiconductor behavior	[87]

**Table 2 biosensors-12-00253-t002:** Summary and comparison of porous nanomaterials enhanced aptasensors.

Material Category	Detailed Electrode Material	Aptamer Sequence	Analytical Method	LOD (fM)	Linear Range	Others	Reference
Hollow and porous nanomaterials	N- GO and Au NPs (on GCE)	TBA1: 5′-SH-(CH_2_)_6_-TTT TTT TTT TTT GGT TGG TGT GGT TGG-3′ TBA2: 5′-SH-(CH_2_)_6_-GGT TGG TGT GGT TGG-3′	DPV	0.027	0.1 fM to 0.1 nM	MnO_2_ nanospheres in a sandwich assay design	[90]
	CuO_2_@aptamer (on Au)	5′-TCT CTC AGT CCG TGG TAG GGC AGG GTT GGG GTG ACT-3′	EIS	330	0.1 to 50 ng mL−1	Cu_2_O Nanospheres	[91]
	PtNPs@Co(II)MOFs@PtNPs (on GCE)	TBA1: 5′-NH_2_-(CH_2_)_6_-GGT TGG TGT GGT TGG-3′ TBA2: 5′-NH_2_-AGT CCG TGG TAG GGC AGG TTG GGG TGA CT-3′	DPV	33	0.1 pM to 50 nM	MOF/Sandwich design	[28]
	Au/hemin@MOFs (on GCE)	5′-NH_2_-(CH_2_)_6_-GGT TGG TGT GGT TGG-3′	DPV	68	0.1 pM to 30 nM	MOF/Sandwich design	[93]
	AuNPs/Ni-MOFs (on GCE)	TBA1: 5′-SH-(CH_2_)_6_-GGT TGG TGT GGT TGG-3′ TBA2: 5′-NH_2_-(CH_2_)_6_-AGT CCG TGG TAG GGC AGG TTG GGG TGA CT- 3′	DPV	16	0.05 pM to 50 nM	MOF/Sandwich design	[94]
Nanochannels	PAA nanochannels as separator	GGT TGG TGT GGT TGG	CV	1000	1 pM to 11.111 nM	PAA with Nanochannels	[97]
	Au NPs coated PET membrane with multiple ion channels	5′-(NH_2_)-(CH_2_)_6_- CCA TCT CCA CTT GGT TGG TGT GGT TGG-3	CV	600	3 to 50 nM	Signal-on mechanism	[98]

**Table 3 biosensors-12-00253-t003:** Summary and comparison of carbon nanomaterial-based aptasensors.

Material Category	Detailed Electrode Material	Aptamer Sequence	Analytical Method	LOD (fM)	Linear Range	Others	Reference
Graphene based	GO (on GCE)	5′-GGT TGG TGT GGT TGG-3′	DPV	7.0 × 10^4^	0.1 nM to 10 nM	Easy and cheap	[103]
	n-Fe_2_O_3/_graphene (on GCE)	5′-GGT TGG TGT GGT TGG-3′	DPV	1000	10 pM to 4.0 nM	Easy immobilization	[104]
	GCE with Porphyrin/graphene (on GCE)	5′- GGT TGG TGT GGT TGG-3′	DPV	2.0 × 10^5^	5 nM to 1.5 μM	Short incubation time	[69]
	Ag NPs/GO (on Au)	TBA1: 5′-SH-(CH_2_)_6_-AGT CCG TGG TAG GGC AGG TTG GGG TGA CT-3′ TBA2: 5′-SH-(CH_2_)_6_-GGT TGG TGT GGT TGG-3′	SWV	3.0 × 10^4^	0.05 nM to 5 nM	Sandwich assay design	[24]
CNT based	SWCNT (on GCE)	5′-AGT CCG TGG TAG GGC AGG TTG GGG TGA CT-3′	CV	1.0 × 10^7^	10 nM to 100 μM	Simple and cheap design	[23]
	TiO_2_-MWCNT/CHIT-SB (on GCE)	5′-AGT CCG TGG TAG GGC AGG TTG GGG TGA CT-3′	DPV	1.0	0.00005 nM to 10 nM	Complex electrode	[111]
	CNT/ZnCr-LDH (on Au)	5′-NH_2_-AGT CCG TGG TAG GGC AGG TTG GGG TGA CT-3′	DPV	0.1	5 fM to 12 nM	Pre-concentration	[112]
	C60/MWCNTs-PEI/PQdot (on SPCE)	5′–NH_2_-AGT CCG TGG TAG GGC AGG TTG GGG TGA CT-3′	DPV	5	50 fM to 20 nM	Large surface area	[45]
Carbon Nanocages	Pt NPs/CNCs (on Au)	5′-SH-(CH_2_)_6_ GGT TGG TGT GGT TGG-3	DPV	10	0.05 pM to 20 nM	Sandwich assay design	[113]

**Table 4 biosensors-12-00253-t004:** Summary and comparison of polymer-based aptasensors.

Material Category	Detailed Electrode Material	Aptamer Sequence	Analytical Method	LOD (fM)	Linear Range	Others	Reference
Polymer-based	CM-PEG-CM (on GCE)	5′-NH_2_-GGT TGG TGT GGT TGG-3′	DPV	15.6	1 pM to 160 nM	Biocompat-ibility and antibiofouling	[63]
	HBPE-CA (on ITO)	5′-NH_2_-GGT TGG TGT GGT TGG-3′	DPV	0.90	10 fM to 100 nM	Whole blood analysis	[22]
	HBPE-SO3 NPs (on GCE)	5′-GGT TGG TGT GGT TGG-3′	DPV	31	2.70 pM to 270 nM	Anticoagulant	[122]
	CSPH (on GCE)	TBA1: 5′-COOH-(CH_2_)_10_-GGT TGG TGT GGT TGG-3′ TBA2: 5′-NH_2_-(CH_2_)_6_-AGT CCG TGG TAG GGC AGG TTG GGG TGA CT-3′	DPV	640	1 pM to 10 nM	Conductive and antifouling	[71]

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
