# Peer review of "Nanomaterial-Based Label-Free Electrochemical Aptasensors for the Detection of Thrombin"

_biosensors, 2022, doi:10.3390/bios12040253_

Round 1

Reviewer 1 Report

The review article titled " Nanomaterial-based Electrochemical Aptasensors for Thrombin Detection " discusses the most recent advances in label-free electrochemical aptasensors for the detection of thrombin, with an emphasis on nanomaterials and nanostructures utilized in sensor design and fabrication. Generally, the manuscript is drafted well although it will need minor revision. I think this paper will be suitable for publication in this journal both in depth and in breadth. Detailed comments are as follows.

  1. Due to the whole text discussed the most recent advances in label-free electrochemical aptasensors for thrombin detection, the title of the manuscript might be “Nanomaterial-based Label-free Electrochemical Aptasensors for the Detection of Thrombin”.
  2. The serial numbers of all headings are incorrect.
  3. To state more clearly and make the review paper more readable, a diagram might be present to illustrate how recent nanotechnological innovations or nanomaterials have aided us to fabricate electrochemical aptasensors for thrombin detection or address the challenges from the detection.
  4. Some related references published most recently need to be cited. Just for example, as stated that nanomaterials with excellent conductivity and intrinsic catalytic activity can therefore act as redox probes without the requirement of additional costly labelling methods or the introduction of various electrodes, the research paper “Journal of Hazardous Materials. 2020, 398: 122778” could be cited and discussed, because it reports a directly assembled nanoarray electrode, which could not only provide excellent conductivity and intrinsic catalytic activity but also provide large specific surface area of the electrode without modification process. 
  5. Figure quality is poor and more illustrations might be needed.

Reviewer 2 Report

The authors present an interesting review regarding the use o sensors for electrochemical detection of thrombin.

Some minor remarks could be addressed as follows:  

Table 1. All the LOD should be given in the same order of magnitude (e.g. pM or fM). The same remark applies for all the tables.

Also, it would beneficial for the readers a short chapter dedicated to electrochemical techniques used in sensing of thrombin and if possible few difficulties which arise by working at such lower concentrations.   

Reviewer 3 Report

The manuscript entitled “Nanomaterial-based Electrochemical Aptasensors for Thrombin Detection” aims to review the most recent advance in label-free electrochemical aptasensors for the detection of thrombin, with an emphasis on nanomaterials and nanostructures utilized in sensor design and fabrication. The performance and advantages/disadvantages of sensors are compared according to the material/structure categories. The manuscript is very well prepared, and the topic is highly important. The Introduction is good and informative. A large amount of data is presented. The discussion is given in-depth and scientifically sound. The literature is up to date. Therefore, I would recommend this manuscript for publication in the present form.

Some small spelling, typing and grammar errors are listed below, corrected and labeled green.

Abstract:

  • …specific selectivity from aptamers with the extraordinary sensitivity from electrochemical techniques and thus have attracted considerable attention…
  • The performance, advantages and limitations of those aptasensors are summarized and compared according to the material structures and compositions.

Introduction:

  • Thrombin is a serine protease involved in hemostasis [1], converting…
  • Thrombin plays a central role in hemostasis as it…
  • In general, the detection of any specific…
  • …suffered from low sensitivity and have therefore not been adopted widely, given the fact that biomarkers of most diseases are present in trace…
  • …to bind their targets for the construction of electrochemical…
  • …miniaturization and mass production makes them promising for…
  • …And lastly, in the case of pathology,…
  • …properties, including biocompatibility, structural stability, and good electrical properties [26]; and more importantly, they can…
  • …intrinsic catalytic activity and can therefore act as redox probes without the requirement of additional costly labeling methods…

Sensor structure and Sensing Principle:

  • …recognition element and the transducer…
  • …can be present in the solution, allowing…
  • This, in turn, generates..
  • ..unique properties, including good…
  • …mainly due to the ability to immobilize th…
  • Figure 1. Conformational changes in aptamer structure due to binding to the analyte. When the electrochemically active probe is attached to the aptamer, binding-induced conformational changes bring the probe either farther (a) or closer (b) to the electrode surface, resulting in signal off or signal on, respectively. Adapted from [36]. In (c), binding of the analyte to aptamer reduces the electrostatic interaction between [Ru(NH3)6]3+ and aptamer, which results in a decrease in the reduction peak. Adapted from [37]. In (d), the binding of
  • Traditionally employed labeled-antibody biosensing [39] requires labels such as enzymes and fluorescent or radioactive molecules to the targeted analyte, and thus is expensive, time-consuming and frequently results in a…
  • Or, more commonly, redox probes can be attached to the aptamer rather than the analyte via…
  • …affinity of the analyte remain un-modified. Upon binding of the analyte to the aptamer, the redox probe is displaced or released due to the aptamer conformation, leading to a change in the electrical signal being measured [18] (Figure 1a-b). Alternatively, redox probes may not be associated with the aptamer, but rather present in the solution, generating the measured signal by diffusing to the electrode [18]. The polyanionic nature of DNA can attract redox active cations, which will be displaced upon the binding of aptamers to thrombin, resulting in a signal output (Figure 1c) [41]. Finally, the binding of…

Structure and Immobilization of Aptamers

  • …which was the first single-stranded DNA oligonucleotide…
  • …as antibody instability and the high cost of this technique…
  • …as an example in Figure 3b. In general, physical adsorption is…
  • …principle could be explained by a simple design…
  • Such a principle could be used in the sandwich assay design of aptasensor…
